# Obesogenic Diet-Induced Neuroinflammation: A Pathological Link between Hedonic and Homeostatic Control of Food Intake

**DOI:** 10.3390/ijms24021468

**Published:** 2023-01-11

**Authors:** José Luis Marcos, Rossy Olivares-Barraza, Karina Ceballo, Melisa Wastavino, Víctor Ortiz, Julio Riquelme, Jonathan Martínez-Pinto, Pablo Muñoz, Gonzalo Cruz, Ramón Sotomayor-Zárate

**Affiliations:** 1Centro de Neurobiología y Fisiopatología Integrativa (CENFI), Instituto de Fisiología, Facultad de Ciencias, Universidad de Valparaíso, Valparaíso 2360102, Chile; 2Escuela de Ciencias Agrícolas y Veterinarias, Universidad Viña del Mar, Viña del Mar 2572007, Chile; 3Programa de Doctorado en Ciencias e Ingeniería para la Salud, Universidad de Valparaíso, Valparaíso 2360102, Chile; 4Programa de Doctorado en Ciencias Mención Neurociencias, Universidad de Valparaíso, Valparaíso 2360102, Chile; 5Escuela de Medicina y Centro de Neurología Traslacional (CENTRAS), Facultad de Medicina, Universidad de Valparaíso, Viña del Mar 2540064, Chile

**Keywords:** diet, feeding control, glia, inflammasome, neuroinflammation, obesity

## Abstract

Obesity-induced neuroinflammation is a chronic aseptic central nervous system inflammation that presents systemic characteristics associated with increased pro-inflammatory cytokines such as interleukin 1 beta (IL-1β) and interleukin 18 (IL-18) and the presence of microglia and reactive astrogliosis as well as the activation of the NLRP3 inflammasome. The obesity pandemic is associated with lifestyle changes, including an excessive intake of obesogenic foods and decreased physical activity. Brain areas such as the lateral hypothalamus (LH), lateral septum (LS), ventral tegmental area (VTA), and nucleus accumbens (NAcc) have been implicated in the homeostatic and hedonic control of feeding in experimental models of diet-induced obesity. In this context, a chronic lipid intake triggers neuroinflammation in several brain regions such as the hypothalamus, hippocampus, and amygdala. This review aims to present the background defining the significant impact of neuroinflammation and how this, when induced by an obesogenic diet, can affect feeding control, triggering metabolic and neurological alterations.

## 1. Introduction

Obesity-induced neuroinflammation is a type of brain aseptic chronic inflammation characterized by high pro-inflammatory cytokines, reactive microglia, and astrogliosis [1,2,3,4]. The intensity of the inflammation induced by obesogenic diets is lower than that caused by infections [5]. However, this neuroinflammation can be evidenced even in normal-weight individuals without insulin resistance or other metabolic disorders [6]. Neuroinflammation induced by a hyperlipidemic diet (lipid, 45 kcal%) was first described in the hypothalamus, evidenced by an increase in c-Jun N-terminal kinase (JNK) and nuclear factor kappa-B (NF-κB) signaling and a reduction in insulin and leptin signaling [1]. Hypothalamic inflammation promotes leptin resistance [7], gliosis, and neuronal death [8]. High-fat diet (HFD)-induced gliosis implicates the activation of astrocytes and microglia [9,10], promoting the release of pro-inflammatory cytokines such as interleukin 1β (IL-1β) and tumor necrosis factor-alpha (TNF-α), resulting in the overexpression of cyclooxygenases and the production of reactive oxygen species [11]. In this work, we review how obesogenic diet-induced neuroinflammation affects the homeostatic and hedonic control of feeding, leading to the perpetuation of obesity and other metabolic and neurological diseases.

## 2. Brain Regulation of Feeding

### 2.1. Homeostatic Control System

This control system is characterized by adapting feeding behavior and energy expenditure to the physiological needs of the energy of body. For example, the behavior associated with a sensory perception is generated under a negative energy balance: hunger. This perception is triggered by stomach movements and stomach rumbling, among others. Conversely, under a positive energy balance, there is no physiological need for energy; thus, satiety is generated, and the behavior of the individual is to avoid eating. The hypothalamus mainly integrates the sensation of hunger/satiety. This diencephalic brain area, on a rostrocaudal axis, is divided into anterior or suprachiasmatic, medial or tuberal, and posterior or mammillary regions; on a medial-lateral axis, the lateral hypothalamus (LH) is recognized as the consuming area of the sensation of hunger. This nucleus receives various interoceptive and exteroceptive information, participating in cognitive, musculoskeletal, autonomic, and endocrine functions [12]. On the other hand, the lateral hypothalamus (LH) is recognized as the satiety center because the electrical activation of this nucleus produces satiety, reducing food intake. In contrast, an electrolyte injury increases food intake [13].

Peripheral hormones, especially ghrelin, leptin, and insulin, participate in the homeostatic control of feeding, mainly by modulating the activity of arcuate nucleus (ARC) neurons [14]. For example, ghrelin (a gastric hormone) is released under the condition of fasting. It triggers the sensation of hunger by activating orexigenic neurons that produce agouti-related peptide (AgRP) and neuropeptide tyrosine (NPY). These AgRP/NPY neurons inhibit anorexigenic paraventricular nucleus (PVN) neurons and activate LH neurons, which behaviorally increase food intake. Moreover, under a positive energy balance, the secretion of hormones such as leptin (produced by white adipose tissue) or insulin (produced by β-pancreatic cells [14]) stimulates the glutamatergic neurons of the VMH that stimulate the anorexigenic neurons that produce the pro-opiomelanocortin (POMC) and cocaine–amphetamine-related (CART) peptides of the ARC [15]. These POMC/CART neurons release the alpha-melanocyte-stimulating hormone peptide α-MSH into the PVN, thereby reducing the food intake. In summary, leptin and insulin are hormones that represent the total amount of energy stored in the body. In contrast, hormone glucagon-like peptide-1 (GLP-1), peptide YY (PYY), and Cholecystokinin (CCK) are hormones released after the intake of food (see Figure 1).

### 2.2. Hedonic Control System 

The reward system, also known as the mesolimbic circuit, regulates the hedonic control of eating. This system consists of dopaminergic neurons of the ventral tegmental area (VTA) that release dopamine (DA) in the nucleus accumbens (NAcc) and prefrontal cortex (PFC) [16]. These neurons exhibit a firing tone regulated by the gamma-aminobutyric acid (GABA) interneurons of the VTA [17,18], which, when inhibited by µ-opioid receptor activation (by endogenous or exogenous agonists), change their firing rate from tonic to phasic [19]. On the other hand, GABAergic (GABA_A_ and GABA_B_ activation in the ventral pallidum) and glutamatergic (NMDA activation in the ventral subiculum) modulations increase the population activity of the VTA DA neurons [20]. Although the agonism of GABA_A_ and GABA_B_ receptors in pedunculopontine decreases the bursting activity of the VTA DA neurons, it is increased by the infusion of the GABA_A_ antagonist [20]. In this context, µ-opioid receptor activation in the NAcc has been shown to promote the intake of food [21]. Dopaminergic neurons in the VTA are activated by drugs of abuse [22] and by natural rewarders such as sex [23] and food [24]. In this context, Di Chiara et al. demonstrated that this dopaminergic activation induced by abuse drugs (e.g., cocaine and amphetamines) [22] or palatable foods (e.g., rich in sugars, fat, and salt) [24] increased the NAcc DA extracellular levels, exerting pleasurable effects in both cases. In addition, psychostimulant drugs that inhibit the DAT (such as cocaine [25], CE-123 (a novel modafinil analog), and R-modafinil [26,27]) increase the extracellular DA levels that are involved in the anorexigenic effect of these drugs. On the other hand, the activation of the D_2_ autoreceptor by enhanced DA levels is also engaged in anorexigenic developments [28]. 

Anatomically, it has been shown that the reward circuit projects towards the NAcc and the olfactory tubercle (OT), an important area related to the olfactory system and the preference for odor-related rewards [29,30]. The OT is a part of the ventral striatum and is innervated from the VTA DA neurons. However, the uptake of DA in the OT is lower than in the NAcc, showing differences in the regulation mediated by the DAT [31].

In addition, peripheral hormones regulating the homeostatic control of feeding have been shown to regulate the hedonic control. This is the case for ghrelin, which activates its receptor, growth secretagogue hormone receptor type 1 GSH-R1, directly activating the dopaminergic neurons in the VTA and their terminals in the NAcc, causing an increase in the DA release [19]. On the other hand, insulin increases the DA reuptake in the VTA by promoting the phosphorylation and activation of the dopamine transporter (DAT) [32]. Orexigenic neurons of the LH that produce the peptide orexin also project their axons on to the VTA, activating the dopaminergic neurons [33]. In this context, Rodrigo A. España et al. demonstrated that the hypocretin 1 receptor antagonist within the VTA reduced a cocaine-induced DA release in the NAcc and reduced the motivation to self-administer sucrose in food-sated, but not food-restricted, rats [34]. Finally, in animal models, a leptin infusion into the VTA was shown to inhibit DA production, decreasing the food intake [24] and reward sensitivity in rats [25] (see Figure 2). In the same context, a mild food restriction augmented the activity of the DA neurons in response to a reward-predicting cue; a pretreatment with leptin abolished this activity [35]. Due to those mentioned above, it is necessary to remember that homeostatic and hedonic regulation systems are interrelated through direct neuronal connections and peripheral hormones. 

### 2.3. Other Changes Induced by a High-Fat Diet

Interestingly, exposure to an HFD affects neurotransmission by affecting the expression of the receptors or other critical proteins in the chemical synapse but also has been shown to increase the expression of inflammatory markers in several brain areas. For example, a few specific hypothalamic changes after a single day of an HFD in rodents led to an increased hypothalamic expression of IL-6 and TNF-α as well as the activation of microglial cells [4]. Exposure to an HFD for three days increased neuroinflammation, gliosis, and the markers of neuronal injuries in rodents [36]. This acute response to a few days of an HFD temporarily receded before returning and becoming chronic upon prolonged HFD feeding [36]. An upregulation of endoplasmic reticulum (ER) stress, a suppressor of cytokine signaling 3 (SOCS3) and the IKKb/NF-kB pathways [37], has been demonstrated as astrogliosis and microgliosis also significantly contribute to the neuroinflammatory tone [3,38]. Strategies to decrease this process in rats normalized several hallmarks of obesity [39]. The cerebral cortex, cerebellum, and brainstem increase pro-inflammatory cytokines and chemokines [40,41] as well as a hippocampal increase in the nuclear factor signaling of cytokines [5,42,43,44,45,46] and also in the amygdala [47,48].

We now review the most relevant aspects of inflammation induced by obesogenic diets and how they affect the brain areas related to feeding control.

## 3. Inflammation

### 3.1. Peripheral Inflammation

Obesity produces a chronic low-grade inflammation within the peripheral tissues; adipose tissue is one of the most sensitive to obesity-induced inflammation [5,49]. In lean individuals, adipose tissue contains multiple immune cells that operate in the T helper 2 (Th2) state, including homeostatic anti-inflammatory macrophages, regulatory T (Treg) cells, type 2 innate lymphoid cells (ILC2), invariant natural killer T (iNKT) cells, natural killer (NK) cells, and eosinophils [50]. In obesity, this immune profile shifts towards a pro-inflammatory state, hallmarked by the proliferation and recruitment of neutrophils, inflammatory macrophages, B cells, cytotoxic T lymphocytes (CD8^+^ T) cells, and T helper (Th) 1 and Th17 cells, along with a reduced abundance of eosinophils, Treg cells, iNKT cells, and ILC2 [50]. Saturated fatty acids directly promote inflammation, facilitating the absorption of lipopolysaccharides (LPS) [51] and activating macrophages, microglia, and astrocytes, similar to LPS by binding to toll-like receptor-4 (TLR4), which triggers NF-κB signaling and promotes cytokine release [52]. In the same context, TNF-α decreases the sensitivity of insulin receptor 1 in adipocytes [53], which can be reversed by the inactivation of TNF-α receptors [54]. Similarly, a TLR4 knockout (KO) demonstrated a reduced preference for fat and sugar intake [55]. TLR4 has a critical role in propagating the activation of the NOD-like receptor protein 3 (NLRP3) inflammasome activated by saturated fatty acids [56].

Another aspect that has recently gained interest is the role of microbiota. In this regard, many publications support the involvement of gut microbiota in the pathophysiology of obesity. In rodent models of diet-induced obesity, gut microbiota modifications were associated with increased intestinal permeability, allowing the passage of food or bacterial antigens that contribute to low-grade inflammation and insulin resistance [57]. The perturbation of the intestinal microbiota and changes in intestinal permeability are considered to be a trigger of inflammation in obesity [58]. In the same sense, metabolic endotoxemia originating from dysbiotic gut microbiota has been identified as a primary mediator for triggering the chronic low-grade inflammation responsible for the development of obesity [59]. Animal studies have demonstrated that gut microbiota could promote adiposity and weight gain by altering the host gene expression, the metabolic and inflammatory pathways, and the gut–brain axis [60].

### 3.2. Central Inflammation or Neuroinflammation

Obesity-induced neuroinflammation was first described in the hypothalamus, evidenced by the upregulation of JNK and NF-κB signaling and a reduced insulin and leptin profile caused by exposure to an HFD [1]. Hypothalamic inflammation leads to leptin resistance [7] and changes in neural projections as well as gliosis and neuronal death [8]. In neuroinflammation, gliosis is characterized by reactive astrocytes and microglia [10,61]. Activated microglia can release various pro-inflammatory cytokines such as IL-1β and TNF-α as well as cyclooxygenases and reactive oxygen species [11]. The deletion of adapter proteins for toll-like receptors protected mice from weight gain and the development of leptin resistance when fed an HFD [37].

#### 3.2.1. Neuroinflammatory Mechanisms

##### Blood-Brain Barrier

The functionality of the BBB depends on a strict architecture. In the ventromedial hypothalamus, the barrier specializes in admitting the dynamic passage of hormones and nutrients from the blood to the energy-sensing arcuate nucleus of the hypothalamus (ARC) and the export of newly synthesized hormones to the pituitary. At the median eminence (ME) level, the barrier has fenestrated capillaries that allow the faster transport of substances into the nutrient-sensing hypothalamic nuclei adjacent to it. However, ME tanycytes, specialized radial glia cells lining the walls of the third ventricle, form a physical barrier to control the correct transport of nutrients and metabolic hormones into the brain parenchyma [62,63,64].

Several mechanisms have been proposed to explain how diet-induced inflammation occurs. One possible mechanism is the alteration of the blood–brain barrier (BBB). A Western Diet (WD) (high fat/high sucrose) consumption increases the BBB permeability, thus allowing immune cell infiltration and leading to hypothalamic inflammation [65]. In this sense, the interactions between the C-reactive protein and the BBB increase the paracellular permeability and induce reactive gliosis [66]. In leptin receptor-deficient (db/db) mice, inflammatory changes in the BBB participated in obesity-related cognitive alterations; rescued cognitive deficits were achieved by reducing the BBB permeability [67]. In this sense, an Evans blue stain entered the central nervous system (CNS) of mice fed with an HFD [68,69,70]; a possible explanation of the mechanism arose from the reduction of tight junction transcripts such as occludin, claudin-5, and claudin-12 in the thalamus and midbrain, increasing the permeability of the BBB in the hippocampus [69,71].

##### Fatty Acids

High-fat and high-sugar diets upregulate the inflammatory NF-κB pathway in the hypothalamus, which is a binding site for regulating energy homeostasis [72]. An inflammatory phenotype was visualized when microglia were treated with saturated fatty acids in vitro [73]. Hypothalamic inflammation contributes to developing and maintaining the obese phenotype; exposure to an HFD for three days produced neuroinflammation, gliosis, and the markers of neuronal injuries in rodents [2]. Furthermore, just one day of an HFD could increase the expression of IL-6 and TNF-α as well as microglial activation [4]. Lipids in the hypothalamus play a potential role in the development of obesity and related metabolic diseases, suggesting that the WD affects lipid accumulation and synthesis in the brain [74], leading to an onset of an increase in inflammatory cytokines, oxidative stress, transcription factor changes, neuron malfunctions, or cell death [75]. A WD drives the inflammatory responses in the hypothalamus, eventually leading to metabolic disorders [76].

#### 3.2.2. Role of Glial Cells in Obesity

Microglia sense their surrounding environment and express pro-inflammatory (M1) or anti-inflammatory (M2) phenotypes [77] in response to the presence of pathogen-associated molecular patterns (PAMPs) [78]. Microglia cells have been proposed to be a critical target of obesity-related inflammation [79]. In this context, systemic injection of lipopolysaccharide (LPS) promoted microglial activation in the hypothalamus [80]. LPS induces a greater expression of the primary histocompatibility complex class 1 (MHC-I), pro-inflammatory cytokines (TNF-α, IL-1, and IL-6), and the activation of cyclooxygenase 1 (COX-1) and NF-κB [81,82]. In a mouse model of diet-induced obesity, a partial substitution of the fatty acid composition of a diet of flax seed oil (rich in C18:3) or olive oil (rich in C18:1) corrected hypothalamic inflammation, which was evaluated by a JNK and NF-κB activity reduction [83]. At the hypothalamic level, metabolic inflammation increased the activation of IKKb/NF-κB up to two times compared to a chronic HFD, and up to five or six times in the case of hyperphagic obese animals [37]. Similarly, microglial cell cultures treated with palmitic acid (a long-chain saturated fatty acid) have a rapid TLR4-dependent microglial activation [84]. Investigations into the ARC observed that microglia activation was critical for altering the energy balance and inducing weight gain during long-term HFD exposure in mice [85], leading to enhanced susceptibility to obesity and a possible suitable pharmacological target. On the other hand, diet-induced microgliosis in the hippocampus has been identified in patients and experimental models of Alzheimer’s disease, providing a potential mechanistic link between obesity/type 2 diabetes and cognitive impairments [86]. A high-caloric diet also increases NLRP3 expression, indicating inflammasome activation and IL-1β production in the hippocampus and amygdala-derived microglia [87] (Figure 3). Interestingly, minocycline (a second-generation tetracycline) is considered to be an inhibitor of microglia-induced neuroinflammation [88], inhibiting intracellular signaling pathways such as p38, ERK1/2, and NF-κB and the release of pro-inflammatory factors, including IL-1β, IL-18, IL-6, and NOS2 [89]. Several studies have shown that treatment with minocycline improves depressive symptoms, decreases the expression of pro-inflammatory cytokines associated with the hyperactivity of the hypothalamic–pituitary–adrenal axis (HPA) [90], and the administration of drugs of abuse in the reward system [91]. Minocycline, an FDA-approved tetracyclic antibiotic with anti-inflammatory properties [92], has been associated with a reduction in HFD-induced weight gain as well as an improvement in insulin sensitivity, a decline in active microglia, and the restoration of alterations in autophagy-related gene networks in the PVN [93]. Microglial activation could influence the energy balance, but the promotion of leptin resistance and impairments in adipose thermogenesis are not yet clear [94]. 

#### 3.2.3. Inflammasomes

The NLRP3 inflammasome is part of the innate immune system activating caspase-1, promoting the release of pro-inflammatory cytokines IL-1β/IL-18 in response to microbial infections and cell damage [95]. At the molecular level, inflammasomes comprise three components: (1) a sensor such as a NOD-like receptor (NLR) or an AIM-2-like receptor (ALR); (2) an apoptosis-associated adapter protein (ASC) containing a caspase recruitment domain; and (3) cysteine inflammatory caspase-1 aspartate [95]. Inflammasomes are also involved in the cleavage of gasdermin-D (GSDM-D), the induction of pyroptosis [96], and obesity-induced inflammation [97]. The NLRP3 inflammasome is one of the most extensively studied inflammasomes [98]. This complex belongs to the nucleotide-binding oligomerization domain-like receptor family (NOD-like) pyrin domain-containing 3 (NLRP3) [97,99]. The NLRP3 inflammasome is activated by molecular patterns associated with cell damage (DAMPS) and favors the proteolytic cleavage of pro-interleukin 1β and 18 through caspase-1, generating the respective active proteins (IL-1β and IL-18), which tend to the inflammatory response [100]. In this context, the NLRP3 inflammasome can be activated by cholesterol crystals and ceramides [101] due to the exacerbation of lipolysis in obesity [102], but can also be activated by bacteria, fungi, and viruses; however, NLRP3 is associated with metabolic and inflammatory conditions such as obesity [103,104]. Unlike other inflammasome complexes, NLRP3 is unique. It mediates the recognition of DAMPs directly involved in cellular metabolism [105]; NLRP3 expression is dependent on NF-κB and, therefore, can be regulated by the components of a diet.

Astrogliosis is characterized by an increase in glial fibrillar acid protein (GFAP), promoting a pro-inflammatory phenotype and hypertrophic morphology. An HFD increases the expression of astrocytic and microglial markers such as GFAP and ionized calcium-binding adapter 1 (Iba-1), respectively, in the hypothalamus [78]. 

## 4. Neuroinflammation and Its Consequences

A considerable amount of information converges on a transversal increase in the inflammatory markers Iba-1, TNF-α, IL-1, IL-6, GFAP, and NF-κB by exposure to obesogenic diets in brain regions related to food intake control such as the hypothalamus, hippocampus, and brainstem [106]. In this sense, it has been described as the increased expression of Iba-1, a microglial activation marker induced by diet [107,108,109], and TNF-α [88,107]. Rats that received a continuous infusion of TNF-α into the third ventricle for three weeks had increased body weight and epididymal adipose tissue, suggesting increased levels of pro-inflammatory cytokine TNF-α in the brain can enhance food intake [6]. In addition, an increase in IL-6 and IL-1β in rodents (mice, rats, and gerbils) that were exposed to obesogenic diets was observed [88,89,90,91,92,93,94,110]. This observation was supported by in vitro studies, where a culture of hypothalamic gonadotropin-releasing hormone (GnRH) neurons released IL-6 in response to exposure to sodium palmitate [111]. GFAP and astrogliosis have been studied in multiple articles [88,89,90,92,109,112,113,114] as well as different experimental animal models such as mice [88,89,90,92,93,109,112,114] and rats [113] fed an HFD [88,89,92,109,112,114] and a WD [113]. All of them coincided with an increase in the expression of GFAP.

## 5. Conclusions

An increase in IL-1β, IL-18, TNF-α, inflammasome NLRP3 activation, astrogliosis, and BBB permeability in the brain areas involved in the control of the intake of food such as the LH and hippocampus are the hallmarks of obesity-induced neuroinflammation. This situation does not act systemically. Instead, it seems to be a differential susceptibility for the inflammatory response. Further analyses are needed to understand the role of the components of neuroinflammation in the pathogenesis of obesity, which will permit a rational exploration of new therapeutic avenues for this worldwide condition.

## Figures and Tables

**Figure 1 ijms-24-01468-f001:**
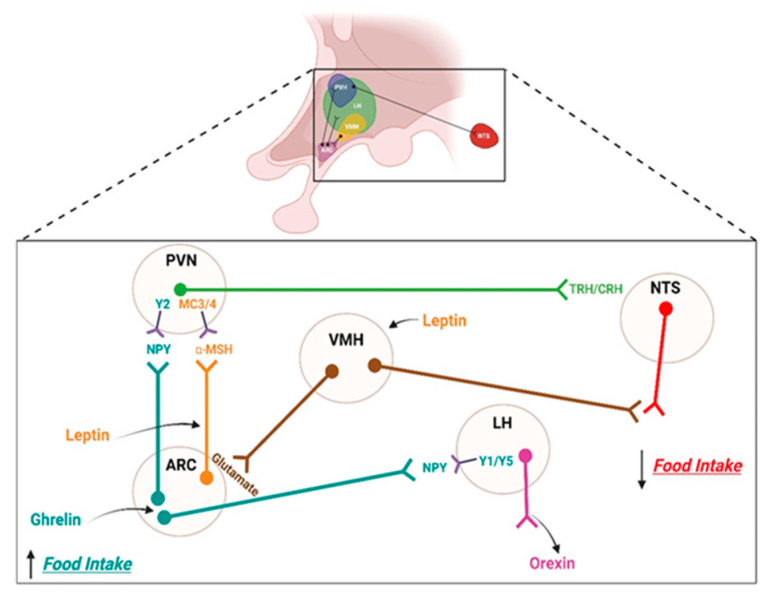
The homeostatic control system of feeding. Increased food intake is triggered by the activation of agouti-related protein (AgRP) and neuropeptide Y (NPY) orexigenic neurons by ghrelin action. These neural activations consequently inhibiting the paraventricular nucleus (PVN) anorexigenic neurons and activating the neurons present in the lateral hypothalamus (LH). On the other hand, hormones such as leptin, which stimulates the glutamatergic neurons of the ventromedial nucleus of the hypothalamus (VMH)—which, in turn, stimulate the anorexigenic pro-opiomelanocortin (POMC) and cocaine–amphetamine-related transcript (CART) neurons of the arcuate nucleus (ARC), releasing the alpha-melanocyte-stimulating hormone α-MSH peptide in the PVN, thus decreasing the food intake—participate in a reduction in the food intake. This figure was created with BioRender.com under a subscription and has a license from BioRender to use the figure in journal publications.

**Figure 2 ijms-24-01468-f002:**
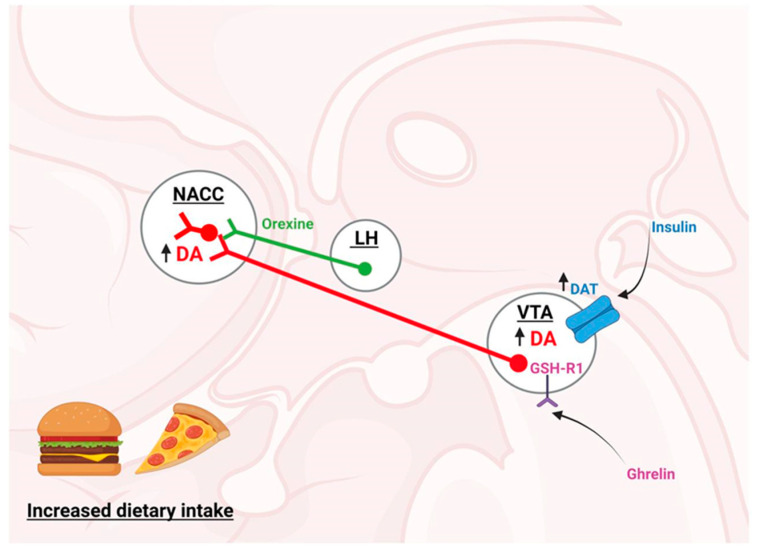
The hedonic control system of feeding. Dietary intake is associated with increased dopamine (DA) availability, mediated by the activation of the secretagogue hormone receptor type 1 (GSH-R1) in the ventral tegmental area (VTA) by ghrelin, thus directly stimulating dopaminergic neurons and their terminals in the nucleus accumbens (NAcc), leading to increased DA release. DA is also re-uptaken in the VTA through dopamine transporter (DAT) phosphorylation, which is increased by the effect of insulin. Finally, it is worth mentioning that the orexigenic neurons of the LH that produce orexin (hypocretin) project their axons on to the VTA, activating the dopaminergic neurons and, consequently, increasing the availability of DA in the environment. This figure was created with BioRender.com under a subscription and has a license from BioRender to use the figure in journal publications.

**Figure 3 ijms-24-01468-f003:**
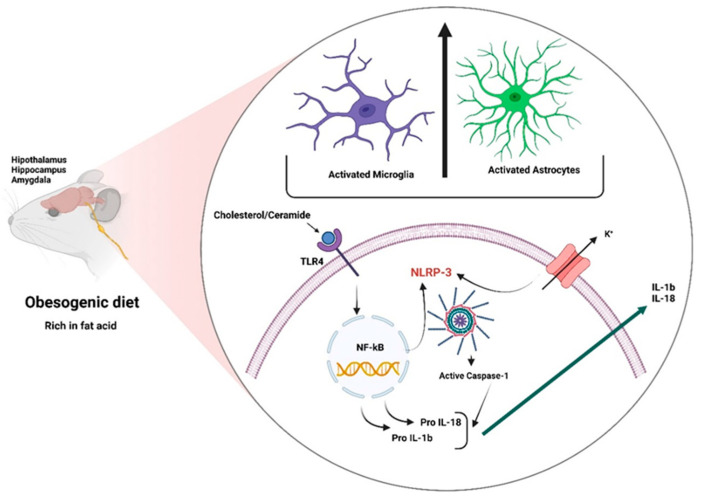
Activating the NLR family pyrin domain-containing 3 (NLRP3) inflammasome signaling pathway requires two signals. Signal 1 or priming is provided by pathogen-associated molecular patterns (PAMPs), danger-associated molecular patterns (DAMPs), cholesterol, or the activation of toll-like receptors (TLRs) or cytokine receptors, leading to the nuclear factor kappa-B NF-κB activation that upregulates the levels of several inflammasome components such as the protein NLRP3, pro-IL-1β, and pro-IL-18. Signal 2 or activation is provided by numerous PAMPs or DAMPs, including viruses, cholesterol, potassium efflux, reactive oxygen species (ROS), extracellular ATP, and lysosomal dysfunctions, among others. ASC, an adaptor protein, recruits NLRP3 and pro-caspase-1 to form the NLRP3 inflammasome complex. Caspase-1 promotes the processing of interleukins for their subsequent release. This figure was created with BioRender.com under a subscription and has a license from BioRender to use the figure in journal publications.

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
