# Peer review of "Obesogenic Diet-Induced Neuroinflammation: A Pathological Link between Hedonic and Homeostatic Control of Food Intake"

_ijms, 2023, doi:10.3390/ijms24021468_

Round 1
Reviewer 1 Report
The manuscript: Obesogenic diet-induced neuroinflammation: a pathological link between hedonic and homeostatic control of food intake, by Marcos et al., is a nice review of the high-fat diet-induced changes in the CNS.
However, the manuscript could be improved significantly.
Remarks and suggestions are as follows:
The first sentence of the abstract has to be altered (Neuroinflammation in general does not have to be aseptic, and so adding "obesity-induced" should correct the statement).
In section 2, an overview of the feeding control systems is given. This section will benefit from listing up-to-now described neuroinflammatory changes in specific nuclei or areas induced by a high-fat diet.
Section 3 needs rearrangement. Sections 3.1 and 3.2 describe peripheral and central inflammation. Peripheral inflammation is again addressed in 3.3.2 (a section that should describe how increased gut permeability affects the BBB). A part about free fatty acids should be more elaborate.
Specific properties of the BBB within the hypothalamic region should be discussed
There is also no reason not to talk about inflammasome and glial cells under 3.2.
The conclusion should be altered in order to reflect the title of the manuscript.
Author Response
REVIEWER Nº1: REPORT FORM
Reviewer 1
Comments and Suggestions for Authors
The manuscript: Obesogenic diet-induced neuroinflammation: a pathological link between hedonic and homeostatic control of food intake, by Marcos et al., is a nice review of the high-fat diet-induced changes in the CNS.
Answer: We appreciate the comment and we will try to respond to each of the comments made by the reviewer
However, the manuscript could be improved significantly.
Remarks and suggestions are as follows:
The first sentence of the abstract has to be altered (Neuroinflammation in general does not have to be aseptic, and so adding "obesity-induced" should correct the statement).
Answer: We appreciate the comment. We have made the correction in the text, highlighting it in yellow (Page 1, line 15).
In section 2, an overview of the feeding control systems is given. This section will benefit from listing up-to-now described neuroinflammatory changes in specific nuclei or areas induced by a high-fat diet.
Answer: We appreciate the comment. We have included a new paragraph that introduces the topic of diet-induced inflammation (Page 4, lines 144 - 161).
Section 3 needs rearrangement. Sections 3.1 and 3.2 describe peripheral and central inflammation. Peripheral inflammation is again addressed in 3.3.2 (a section that should describe how increased gut permeability affects the BBB). A part about free fatty acids should be more elaborate.
Answer: We appreciate the comment. Section 3 has been fixed, describing the role of BBB and fatty acids more clearly (Pages 5 – 8 Lines 163 - 318).
Specific properties of the BBB within the hypothalamic region should be discussed
Answer: We appreciate the comment. This topic has been included in sub-section 3.2.1.1. (Page 6, lines 210 - 218).
There is also no reason not to talk about inflammasome and glial cells under 3.2.
Answer: We appreciate the comment and was fixed (Pages 7 - 8, lines 282- 318).
The conclusion should be altered in order to reflect the title of the manuscript.
Answer: We appreciate the comment and was fixed (Page 8, lines 320 - 335).
Reviewer 2 Report
Dear Authors, this review is interesting to the field and generally well written. Please find below some concerns according to my experience.
- Please check carefully for too-long sentences, to be possibly splitted. For example: lines 23-26 / 206-211
- Introduction
Lines 96-97. Given the importance of olfaction in food intake and associated reward, it is worth mentioning VTA-DA projections to the olfactory tubercle (a part of the ventral striatum together with the NAc). Please find possible sources:
Ikemoto S. Dopamine reward circuitry: two projection systems from the ventral midbrain to the nucleus accumbens-olfactory tubercle complex. Brain Res Rev. 2007 Nov;56(1):27-78. doi: 10.1016/j.brainresrev.2007.05.004.
hang Z, et al,. Activation of the dopaminergic pathway from VTA to the medial olfactory tubercle generates odor-preference and reward. Elife. 2017 Dec 18;6:e25423. doi: 10.7554/eLife.25423.
Wakabayashi KT, et al. Application of fast-scan cyclic voltammetry for the in vivo characterization of optically evoked dopamine in the olfactory tubercle of the rat brain. Analyst. 2016 Jun 21;141(12):3746-55. doi: 10.1039/c6an00196c.
Line 97-99. Please consider also Glutamate in the “tonic to phasic” switch of the pattern, ad reviewed for example in:
Floresco SB, et al. Afferent modulation of dopamine neuron firing differentially regulates tonic and phasic dopamine transmission. Nat Neurosci. 2003 Sep;6(9):968-73. doi: 10.1038/nn1103.
Line 101. Di Chiara and Imperato shows dopamine “system/release” activation by microdialysis, that is dopamine levels in projection areas. Indeed, dopamine neuron “firing” in the VTA (as you mentined) can be either inhibited by drugs of abuse, tipically by DAT ihibitors such as cocaine (Einhorn LC, et al (1988): Electrophysiological effects of cocaine in the mesoaccumbens dopamine system: Studies in the ventral tegmental area. J Neurosci 8:100 –112.)
or new psychostimulants
Sagheddu C, et al. Neurophysiological and Neurochemical Effects of the Putative Cognitive Enhancer (S)-CE-123 on Mesocorticolimbic Dopamine System. Biomolecules. 2020 May 18;10(5):779. doi: 10.3390/biom10050779.
Avelar AJ, et al. Atypical dopamine transporter inhibitors R-modafinil and JHW 007 differentially affect D2 autoreceptor neurotransmission and the firing rate of midbrain dopamine neurons. Neuropharmacology. 2017 Sep 1;123:410-419. doi: 10.1016/j.neuropharm.2017.06.016.)
possibly via D2 autoreceptor agonism by enhanced dopamine (Ford CP. The role of D2-autoreceptors in regulating dopamine neuron activity and transmission. Neuroscience. 2014 Dec 12;282:13-22. doi: 10.1016/j.neuroscience.2014.01.025.)
Given the anorexigenic effect of psychostimulants, such mechanism should be clearly stated (and cited) in this review.
Lines 117-119. Please cite specifically van der Plasse, G.,. et al. Modulation of cue-induced firing of ventral tegmental area dopamine neurons by leptin and ghrelin. Int J Obes 39, 1742–1749 (2015). https://doi.org/10.1038/ijo.2015.131 beyon reviews
- Section 3.3.2 about Gut Microbiota would deserve more details and citations given specific association with food processing
- Section 3.4: sounds confused. Please consider extensive editing for a more linear presentation of the topic. For example the acronym is introduced after 9 lines (moreover, it was already introduced in the line 150).
- Please consider to change the title in Section 5. Indeed, several subsections about neuroinflammation were previously included in the Section 3, therefore this sounds unclear.
Author Response
REVIEWER Nº2: REPORT FORM
Reviewer 2
Dear Authors, this review is interesting to the field and generally well written. Please find below some concerns according to my experience.
Answer: We appreciate the positive comment of the reviewer.
- Please check carefully for too-long sentences, to be possibly splitted. For example: lines 23-26 / 206-211
Answer: We appreciate the comment of the reviewer. We fixed the sentences (Page 1, lines 22 - 26 ; Page 7, lines 282 - 287).
- Introduction
Lines 96-97. Given the importance of olfaction in food intake and associated reward, it is worth mentioning VTA-DA projections to the olfactory tubercle (a part of the ventral striatum together with the NAc). Please find possible sources:
- Ikemoto S. Dopamine reward circuitry: two projection systems from the ventral midbrain to the nucleus accumbens-olfactory tubercle complex. Brain Res Rev. 2007 Nov;56(1):27-78. doi: 10.1016/j.brainresrev.2007.05.004.
- Zhang Z, et al, Activation of the dopaminergic pathway from VTA to the medial olfactory tubercle generates odor-preference and reward. Elife. 2017 Dec 18;6:e25423. doi: 10.7554/eLife.25423.
- Wakabayashi KT, et al. Application of fast-scan cyclic voltammetry for the in vivo characterization of optically evoked dopamine in the olfactory tubercle of the rat brain. Analyst. 2016 Jun 21;141(12):3746-55. doi: 10.1039/c6an00196c.
Answer: We appreciate the comment of the reviewer. We include a new paragraph (Page 3, lines 111 - 115).
Line 97-99. Please consider also Glutamate in the “tonic to phasic” switch of the pattern, ad reviewed for example in:
- Floresco SB, et al. Afferent modulation of dopamine neuron firing differentially regulates tonic and phasic dopamine transmission. Nat Neurosci. 2003 Sep;6(9):968-73. doi: 10.1038/nn1103.
Answer: We appreciate the comment of the reviewer. We include a new paragraph (Page 3, lines 96 - 100).
Line 101. Di Chiara and Imperato shows dopamine “system/release” activation by microdialysis, that is dopamine levels in projection areas. Indeed, dopamine neuron “firing” in the VTA (as you mentioned) can be either inhibited by drugs of abuse, tipically by DAT inhibitors such as cocaine (Einhorn LC, et al (1988): Electrophysiological effects of cocaine in the mesoaccumbens dopamine system: Studies in the ventral tegmental area. J Neurosci 8:100 –112.)
- or new psychostimulants
- Sagheddu C, et al. Neurophysiological and Neurochemical Effects of the Putative Cognitive Enhancer (S)-CE-123 on Mesocorticolimbic Dopamine System. Biomolecules. 2020 May 18;10(5):779. doi: 10.3390/biom10050779.
- Avelar AJ, et al. Atypical dopamine transporter inhibitors R-modafinil and JHW 007 differentially affect D2 autoreceptor neurotransmission and the firing rate of midbrain dopamine neurons. Neuropharmacology. 2017 Sep 1;123:410-419. doi: 10.1016/j.neuropharm.2017.06.016.)
- possibly via D2 autoreceptor agonism by enhanced dopamine (Ford CP. The role of D2-autoreceptors in regulating dopamine neuron activity and transmission. Neuroscience. 2014 Dec 12;282:13-22. doi: 10.1016/j.neuroscience.2014.01.025.)
Given the anorexigenic effect of psychostimulants, such mechanism should be clearly stated (and cited) in this review.
Answer: We appreciate the comment of the reviewer. We include a new paragraph (Page 3, lines 106 - 110).
Lines 117-119. Please cite specifically van der Plasse, G.,. et al. Modulation of cue-induced firing of ventral tegmental area dopamine neurons by leptin and ghrelin. Int J Obes 39, 1742–1749 (2015). https://doi.org/10.1038/ijo.2015.131 beyon reviews
Answer: We appreciate the comment of the reviewer. We added this reference in concordance with the advice (Page 4, lines 138 – 140).
- Section 3.3.2 about Gut Microbiota would deserve more details and citations given specific association with food processing
- Section 3.4: sounds confused. Please consider extensive editing for a more linear presentation of the topic. For example the acronym is introduced after 9 lines (moreover, it was already introduced in the line 150).
- Please consider to change the title in Section 5. Indeed, several subsections about neuroinflammation were previously included in the Section 3, therefore this sounds unclear.
Answer: We appreciate the comments of the reviewer and were fixed.
Reviewer 3 Report
This review is about the neuroinflammatory effects induced by obesogenic diets. It is interesting and overall well written. I suggest authors check some matches such as at line 65, reference (13), and at line 97 reference (16).
Author Response
REVIEWER Nº3: REPORT FORM
Reviewer 3
Comments and Suggestions for Authors
This review is about the neuroinflammatory effects induced by obesogenic diets. It is interesting and overall well written. I suggest authors check some matches such as at line 65, reference (13), and at line 97 reference (16).
Answer: We appreciate the positive comment of the reviewer. We check the reviewer's suggestion).
Round 2
Reviewer 2 Report
Dear Authors,
I appreciated the effort to improve the manuscript.
Kind regards
Author Response
Dear Reviewer
We appreciate your positive review and comments that improved our manuscript.
Ramón Sotomayor-Zárate